# Potential of Curcumin in Skin Disorders

**DOI:** 10.3390/nu11092169

**Published:** 2019-09-10

**Authors:** Laura Vollono, Mattia Falconi, Roberta Gaziano, Federico Iacovelli, Emi Dika, Chiara Terracciano, Luca Bianchi, Elena Campione

**Affiliations:** 1Dermatology Unit, Department of “Medicina dei Sistemi”, University of Rome Tor Vergata, Via Montpellier, 1–00133 Rome, Italy; 2Department of Biology, University of Rome “Tor Vergata”, Via della Ricerca Scientifica, 1–00133 Rome, Italy; 3Microbiology Section, Department of Experimental Medicine, University of Rome Tor Vergata, Via Montpellier, 1–00133 Rome, Italy; 4Dermatology Unit, Department of Experimental, Diagnostic and Specialty Medicine-DIMES, University of Bologna, Via Massarenti, 1–40138 Bologna, Italy; 5Neurology Unit, Guglielmo de Saliceto Hospital, 29121–29122 Piacenza, Italy

**Keywords:** curcumin, antioxidants, molecular docking, inflammatory skin diseases, psoriasis, atopic dermatitis, iatrogenic dermatitis, wound care, skin aging, inflammaging, skin cancer, skin infections

## Abstract

Curcumin is a compound isolated from turmeric, a plant known for its medicinal use. Recently, there is a growing interest in the medical community in identifying novel, low-cost, safe molecules that may be used in the treatment of inflammatory and neoplastic diseases. An increasing amount of evidence suggests that curcumin may represent an effective agent in the treatment of several skin conditions. We examined the most relevant in vitro and in vivo studies published to date regarding the use of curcumin in inflammatory, neoplastic, and infectious skin diseases, providing information on its bioavailability and safety profile. Moreover, we performed a computational analysis about curcumin’s interaction towards the major enzymatic targets identified in the literature. Our results suggest that curcumin may represent a low-cost, well-tolerated, effective agent in the treatment of skin diseases. However, bypass of limitations of its in vivo use (low oral bioavailability, metabolism) is essential in order to conduct larger clinical trials that could confirm these observations. The possible use of curcumin in combination with traditional drugs and the formulations of novel delivery systems represent a very promising field for future applicative research.

## 1. Introduction

Curcumin is a bright yellow chemical compound isolated from *Curcuma longa* L. (turmeric) plants (Zingiberaceae) [1]. Turmeric has been historically used in herbalism as a traditional medical remedy for cutaneous and gastrointestinal inflammation, weight control, and poor digestion [2,3,4].

Recently, conventional medicine is directing a lot of effort towards identifying novel, low-cost, safe molecules that may be used in the treatment of inflammatory, neoplastic, and infectious diseases. Numerous in vitro and in vivo studies have examined curcumin’s anti-inflammatory, anticancer, and antimicrobial properties, both individually and combined with traditional treatments. This paper aims to provide an overview on the current knowledge regarding curcumin’s effects on skin conditions alongside with its bioavailability and safety profile through the analysis of the most relevant studies published to date, providing suggestions for further research (Figure 1). Molecular docking studies describing the interaction of curcumin with molecular targets involved in the development of skin disorders are nowadays not available in literature. We therefore complemented our data with original results, obtained through molecular docking analysis, regarding curcumin’s binding mode and interaction towards six major enzymatic targets, indicated in this review as responsible for several dermatological conditions. 

### 1.1. Bioavailability of Curcumin

According to Nutraceutica Bioavailability Classification Scheme (NuBACS), curcumin shows poor bioaccessibility, due to its low solubility in water and low stability [5]. Curcumin also undergoes extensive first-pass metabolism through its glucuronidation and sulfation, with the production of metabolites that have shown to have significant lower biological activities compared to parent curcumin and that are rapidly eliminated [6]. A curcumin-converting enzyme named “NADPH-dependent curcumin/dihydrocurcumin reductase” (CurA) has been purified from *Escherichia Coli*, shedding new light on the role of human intestinal microorganisms in the mechanism of curcumin metabolism in vivo [7]. Preclinical and clinical studies assessed that curcumin is poorly absorbed following oral administration. In rats, only 60% of the dose of curcumin administered orally was adsorbed, with negligible quantities (<20 μg/tissue) detected in liver and kidney from 15 min up to 24 h after administration albeit 38% of the initial dose being detected in the large intestine and patients taking curcumin orally show plasmatic concentration of the compound at nanomolar levels, with limited biological effects [8,9,10,11,12]. To overcome this limitation, combination with adjuvant substances such as piperine, encapsulation with polylactic-co-glycolic acid (PLGA) and cyclodextrin (CD), or formulation in liposome, micelles, nanoparticles, nanomicellizing solid dispersion based on rebaudioside A and dispersion with colloidal submicron-particles have been recently proposed, showing to enhance curcumin bioavailability and therapeutic potential [6,13,14,15,16,17,18]. We present some significant results induced by curcumin administered in several formulations below in this review (Table 1). Ongoing clinical trials investigating the topical or systemic use of curcumin in skin conditions are listed in Table 2.

Intravenous use of curcumin has been proposed in order to improve curcumin bioavailability and increase its efficacy. Serum curcumin levels after intravenous administration were significantly higher than the one observed after oral administration in rats [36,40]. In animal models, curcumin infusion showed significant anticancer effects without inducing toxicity [6,41]. A randomized, placebo-controlled double-blind phase I dose escalation study investigated the pharmacokinetics, safety, and tolerability of short-term intravenous administration of liposomal curcumin in healthy subjects with good results in terms of bioavailability and safety [37]. Pharmacokinetics of curcumin infusion seems to depend on co-medication and health status, as highlighted by a recent clinical study [38]. However, these interesting albeit limited data must be confirmed by larger clinical trials with longer follow-up in order to recommend this route of administration.

Curcumin showed a good accessibility and bioactivity when administered topically, especially when incorporated in novel formulations such as chitosan-alginate sponges, polymeric bandages, alginate foams, collagen films, nano-emulsion, hydrogel, and β-cyclodextrin-curcumin nanoparticle complex, making curcumin eligible as a therapeutic agent for the topical treatment of skin conditions [21,22,23,24,25,26,27,42]. Investigating the possible interactions between curcumin and other chemicals commonly used in topical skin treatments may provide useful insights for the development of new effective combination preparations, tailored for different conditions. 

### 1.2. Curcumin’s Safety Profile

Curcumin is recognized as a safe compound by Food and Drug Administration (FDA). Numerous preclinical and clinical studies assessed the safety of this compound [43,44,45]. The maximum recommended dose varies, ranging from a maximum daily intake of 3 mg/kg to 4–10 g [46]. In a clinical study, curcumin was not detected in the serum of healthy subjects administered up to 8000 mg/day, and only low levels were detected in two subjects administered 10,000 or 12,000 mg. No harmful effect was observed in any of the subjects, regarding a daily intake of 12,000 mg as safe in healthy individuals [47]. 

A good safety profile of curcumin was observed also in patients with cardiovascular risk factors and patients affected by high risk conditions or pre-malignant lesions of internal organs taking a dose of curcumin ranging from 500 to 8000 mg/day for 3 months [43,48]. This safety has been observed also in patients with advanced colorectal cancer taking a dose of curcumin ranging from 36 to 180 mg/day for up to 4 months, in breast cancer patients undergoing radiotherapy while taking up to 6000 mg/day of curcumin, and advanced pancreatic cancer patients taking 8000 mg/day of curcumin for 2 months [49,50,51].

Other studies in both healthy subjects and patients affected by several conditions such as advanced colorectal cancer, cholangitis and ulcerative colitis reported mild and manageable gastrointestinal symptoms with a daily intake of up to 8000 mg of curcumin [52,53,54,55]. Alongside these data, a minority of patients affected by primary sclerosing cholangitis taking up to 1400 mg/day of curcumin reported only mild symptoms such as headache or nausea [56]. Controversially, intractable abdominal pain after assumption of curcumin at a dose of 8000 mg/day has also been reported in patients affected by advanced pancreatic cancer taking gemcitabine [57]. It may be speculated whether curcumin-induced COX inhibition and the subsequent inhibition of prostaglandin (PG) synthesis (see below) plays a role in the development of gastrointestinal side effects in patients suffering other gastrointestinal conditions. However, no sound explanation is available to date. 

Short-term intravenous dosing of liposomal curcumin has been indicated as safe up to a dose of 120 mg/m in a clinical trial on healthy subjects, whereas in a dose escalation study on metastatic cancer patients a dose of 300 mg/m^2^ over 6 h appeared to be the maximum tolerated dosage [37,39]. However, changes in red blood cell morphology may represent a dose limiting sign of toxicity, and one case of hemolysis and one death associated with intravenous curcumin preparation were reported, suggesting the need for further data regarding the safety and recommended dosages of curcumin administered intravenously [36,39,58]. 

It is worth mentioning that the majority of studies assessing curcumin safety profile has been conducted for short periods of time. No sound evidence is available to date regarding the consequences of long-term use of this compound. Although doses recommended for over-the-counter curcumin are generally lower than the ones in clinical studies mentioned above, supplements containing this compound are widely available to the general public and are increasingly popular. In this regard, recent reports of liver diseases related to curcumin assumption drove the medical community’s attention to the possible liver toxicity of this molecule [59]. The exact role of curcumin in the development of these conditions still has to be elucidated, and a possible contamination of supplements with lead has been postulated. Until further data is available, surveillance is needed, especially in long-term use, in the over-the-counter context and in patients affected by liver conditions. 

### 1.3. Curcumin for the Treatment of Psoriasis 

Psoriasis (PsO) is a chronic inflammatory, multisystemic, and multifactorial disease affecting about 3% of the world population. The clinically observed thick, silvery plaques are the result of uncontrolled proliferation of keratinocytes. The first step in psoriasis pathogenesis is the activation of mature and inflammatory dendritic cells (DC), leading to hyperproduction of proinflammatory molecules such as cytokines, chemokines and antimicrobial peptides (AMPs). Cytokines belonging to the IL-23/T-helper-(Th)-17 axis and type I interferons (IFNs) play a paramount role, being a target of several monoclonal antibodies used in the treatment of psoriasis [60]. 

Curcumin is able to suppress the excessive production of TNF-α by activated macrophages [61,62,63,64,65]. Curcumin has been shown to directly bind to the receptor-binding sites of TNF-α by covalent and non-covalent interactions, blocking the subsequent TNF-dependent activation of NF-κB [66,67]. It has been also observed that curcumin can inhibit a TNF-α promoter by its methylation and is able to impair lipopolysaccharide (LPS) signaling, responsible of the induction of TNF-α production, by acting on toll-like receptors (TLRs) 2 and 4 [68,69,70]. Moreover, curcumin is a non-competitive inhibitor of Phosphorylase kinase (PhK), a serine/threonine-specific protein kinase. Levels of PhK in human skin samples taken from patients affected by untreated active psoriasis, resolving psoriasis undergoing topical treatment, and non-psoriatic subjects showed to directly correlate to the activity of psoriasis. In this study, decreased levels of PhK in samples of plaques treated with curcumin 1% alcoholic gel as well as other traditional topical treatment were associated with decreased keratinocyte transferrin receptor (TRR) expression, severity of parakeratosis, and density of epidermal CD8+ T cells [71]. These preliminary observations may suggest that agents capable to inhibit PhK activity, such as curcumin, could be considered suitable candidates the topical treatment of psoriasis [72,73]. 

In animal studies, daily applications of 1% curcumin gel reduced skin psoriasis-like inflammation artificially induced by imiquimod, through the inhibitions of the potassium channels (subtypes Kv1.3) expressed in T cells and the reduction of IL-17A, IL-17F, IL-22, and other pro-inflammatory cytokines in ear samples taken from mice [74,75]. Clinically, daily applications of a turmeric tonic significantly reduced the cutaneous symptoms and quality of life of patients affected by scalp psoriasis compared to the placebo [76]. A recent randomized, double-blind, placebo-controlled clinical trial reported the anti-psoriatic effects of oral administration of Meriva, a novel bioavailable lecithin-based delivery form of curcumin, observing a reduction of cutaneous symptoms together with a decrease of serum levels of IL-22. Furthermore, the treatment increased the anti-psoriatic effects of topical steroids in these patients when treated in combination [19]. 

Curcumin oral administration (40 mg/kg, for 20 days) resulted in significant reduction of the serum levels of IL-2, IL-12, IL-22, IL-23, IFN-gamma, and TNF-alpha in psoriatic mice, reducing psoriasis-associated inflammation as well as hyper-proliferation of keratinocytes [77]. Clinically, a phase II clinical trial confirmed the efficacy of oral curcumin on cutaneous symptoms of plaque psoriasis, reporting an excellent safety profile [67]. Interestingly, a double blind, placebo-controlled randomized clinical trial reported that oral administration of curcumin formulated as nanoparticles potentiated the effectiveness of acitretin in psoriatic patients and resulted in control of their serum cholesterol levels, suggesting a role of this compound as adjuvant treatment in moderate-to-severe psoriasis [78].

### 1.4. Curcumin for the Treatment of Atopic Dermatitis

Atopic dermatitis (AD) is a chronic, pruritic inflammatory skin disease of unknown etiology, resulting from a complex interplay between genetic, environmental, and immune factors [79]. It usually starts in early infancy, but also affects a substantial number of adults. The prevalence of atopic diseases has increased abruptly in recent years in most Westernized societies, resulting in considerable research into safe, economically viable, and readily manufactured therapies for AD [80].

An imbalance in the T cell subsets is crucial in the pathogenesis of AD. The early stages are characterized by an abnormal production of cytokines such as IL-4, IL-5, IL-13, and IL-31 by Th2, whereas in later phases a switch from the initial Th2 response to a Th1 type-immune response is observed, with excessive release of IL-1, IL-6, TNF-α, IL-12, and IL-18 by recruited monocytes [81].

In Asian countries, curcumin has been traditionally used to manage atopic dermatitis symptoms [82].

The phytocomponent *p*-hydroxycinnamic acid (HCA) isolated from *Curcuma longa* has been shown to modulate the protein kinase C (PKC) theta (PKCθ) pathway in vitro, leading to the inhibition of T-cell activation [83]. In animal study, oral administration of HCA induced a reduction in the production of proinflammatory cytokines by keratinocytes in both the ear tissues and in vitro, improving cutaneous signs of AD such as dermo-epidermal thickening and inflammation in mice [71]. 

Clinically, a combination herbal extract cream (Herbavate^®^) containing *C. longa* applied daily alleviated erythema, scaling, thickening, and itching in patients affected by eczema [84]. However, the design of the study (non-comparative study, lack of control group, high drop-out rate, impossibility to distinguish between the effects of turmeric and the other cream components) makes the significance of the results debatable. Further randomized, comparative clinical trials are needed in order to establish the potential role of curcumin in the treatment of AD.

Contact dermatitis and contact urticaria after topical application of curcumin-based creams have been reported [85,86,87]. Once more, surveillance is advisable in highly reactive patients, such as the ones affected by atopic dermatitis.

### 1.5. Curcumin for the Treatment of Iatrogenic Dermatitis

Iatrogenic dermatitis is a non-specific term used to indicate a variety of inflammatory skin conditions directly attributable to medical procedures or drug administration. Radiation-induced dermatitis developing in patients undergoing radiotherapy sessions and allergic contact dermatitis due to applied medicaments are typical examples of iatrogenic dermatitis. 

Several studies propose curcumin as a natural, safe, widely available, and inexpensive treatment for the management of iatrogenic dermatitis. 

In an animal model, daily topical application of curcumin showed to improve epithelial cell survival and recovery in irradiated skin, reducing the expression of cyclooxygenase-2 and nuclear factor-kappaB [88].

In a randomized, double-blind, placebo-controlled clinical trial oral curcumin administration (6 g/day) during radiotherapy sessions reduced the severity of radiation-induced dermatitis in 30 breast cancer patients [50].

Oral administration of curcumin (4 g/day) showed to prevent capecitabine-induced hand-foot syndrome (HFS) in 40 cancer patients undergoing treatment with capecitabine, with no toxicity associated with curcumin. Interestingly, no correlations between inflammatory parameters such as IL-6, TNF-α, neutrophil/lymphocyte index, and HFS severity was found, thus the mechanism behind this preventive effect is not fully elucidated [89].

In addition, a placebo-controlled study reported that oral administration of curcumin (1 gr/day) combined with piperine for 4 weeks improved sulphur mustard-induced chronic pruritic symptoms and DLQI of 46 patients compared with placebo. The authors observed a significant reduction in the levels of various inflammatory markers such as IL-8, hs-CRP CGRP in the patients receiving curcumin compared with placebo, and a concurrent reduction of substance *p* (*p* < 0.001) as well as significant elevations in serum superoxide dismutase (SOD), glutathione peroxidase (GPx) and catalase activities, further confirming the well-documented antioxidant activities of curcumin (discussed below). The authors state that the abovementioned effects may have been influenced by the association of curcumin with piperine, a well-documented bioavailability enhancer [90].

### 1.6. Curcumin for Wound Care

Wound treatment represents a therapeutic challenge with significant economic impact on healthcare systems worldwide, with its cost rising sharply [26]. 

Wound healing is a complex, dynamic process that involves a sequence of cellular and molecular events. It can be divided in a simplified manner into three phases: (1) hemostasis and inflammation, (2) proliferation with formation of granulation tissue, and (3) remodeling, with formation of new epithelium and scarring [91].

During the inflammatory phase a significant number of neutrophils are recruited at the wounded site, releasing proteases, reactive oxygen species (ROS), and inflammatory mediators such as TNF-α and IL-1 [92,93]. As mentioned above (see Section 1.3) curcumin is able to reduce inflammation through the inhibition of nuclear factor κB (NF-κB) and the suppression of TNF-α expression, as well as through the impairment of LPS signaling. Moreover, curcumin exerts its anti-inflammatory effects by acting on other signaling pathways, such as peroxisome proliferator-activated receptor-gamma (PPAR-γ) and myeloid differentiation protein 2-TLR 4 co-receptor (TLR4-MD2) [94,95,96,97]. Excessive oxidative stress plays a major role in prolonged inflammation, a significant feature in the pathogenesis of chronic non-healing wound [98,99]. In fact, while low levels of ROS are physiologically formed during the physiologic wound healing process, their excessive production cannot be balanced by the cellular antioxidant system, leading to oxidative stress, lipid peroxidation (LPx), DNA breakage and enzyme inactivation, including free-radical scavenging enzymes, in a self-perpetuating cycle resulting in chronic disease [100]. The reducing potential of its electron-donating groups allows curcumin to restore the redox balance and suppress transcription factors related to oxidation, while sustaining the production and activity of antioxidant enzymes and their constituents, such as glutathione (GSH) [24,101,102,103,104,105]. Moreover, a protective action of curcumin against hydrogen peroxide has been observed in vitro in human keratinocytes and fibroblasts [106]. 

During the proliferative phase of wound healing, the dermis is invaded by proliferating fibroblasts producing immature ECM proteins (EDA fibronectin and type III collagen) as well as activating growth factors such as TGF-β1, leading to reparation of the wounded dermal layer [107,108,109,110,111,112,113]. 

Simultaneously, keratinocytes migrate at the wounded site, where they proliferate and differentiate in order to restore the overlying epithelium [114,115]. A major role in this process is played by hair follicle stem cells [116,117].

Curcumin may exert significant action during the proliferative phase [118]. In fact, it has been demonstrated that curcumin is able to reduce the number of membrane matrix metallo-proteinases (MMPs), increase the hydroxyproline and collagen synthesis, and accelerate the maturation of collagen fibers [24,119]. In addition, curcumin also promotes the differentiation of fibroblasts into myofibroblasts, which marks the beginning of wound contraction, and reduces the epithelization period in wounds [119,120,121,122,123,124].

In animal models, daily curcumin topical application accelerated wound healing in irradiated mini-pigs, and the application of chrysin-curcumin-loaded nanofibers reduced the levels of IL-6, MMP-2, TIMP-1, TIMP-2, and iNOS gene expression in male rats, resulting in the acceleration of the healing process of surgical wounds [28,88]. Transdermally applied curcumin on surgical wounds on rats produced marked inhibition of H_2_O_2_-induced damage to keratinocytes and fibroblasts, while application of curcumin-oligochitosan nanoparticle complex or with application of oligochitosan coated curcumin-loaded-liposomes resulted in faster healing of surgical wounds in mice compared with controls [29,30]. In a diabetic rat model, wounds treated with curcumin showed an accelerated reepithelization rate compared with untreated controls [125]. Treatment with curcumin-loaded polymeric bandages resulted in significantly lower expression of PI3K and pAKT, indicative of an inhibition of the PI3K/AKT/NFκB axis, reduced LPx levels, and increase in collagen compared with controls.

Clinically, patients affected by diabetic wounds treated with curcumin loaded chitosan nanoparticles impregnated into collagen-alginate scaffolds reported a significantly faster healing process compared to those treated with patients receiving treatment with placebo scaffold [35].

As mentioned above, topical application of curcumin seems to have more pronounced effects on wound healing compared to its oral administration in the treatment of wounds, owing to higher accessibility of the drug at the wound site [94,125,126,127]. Many new formulations of curcumin have been developed in order to achieve better topical application at the wound site, such as chitosan-alginate sponges, curcumin-loaded polymeric bandages, alginate foams, collagen films, and nano-emulsion and hydrogel [26,27,28,29,30,31]. The incorporation into these formulations resulted in increased curcumin bioactivity, although no formulation showed a significant difference in its effect compared to the others. However, formulation as nanoparticles seems to be of special interest, as it increases curcumin bioavailability and half-life and enhances its water dispersibility [27,30,31,32]. Further studies comparing nanoparticles with other formulations are needed in order to confirm these observations.

### 1.7. Curcumin for the Treatment of Skin Aging: The “Inflammaging” Issue

Human aging is a very complex process that occurs in an intricate biological and physiological setting, depending on a complex interaction between genetic, environmental, and stochastic factors. The term “exposome” has been proposed to describe the totality of exposures to which an individual is subjected from conception to death, including both external and internal factors as well as the human body’s response to these factors. Specifically, not clinically evident infections, sun radiations (UVA and UVB), air pollution, and tobacco smoke have been listed as environmental factors [128]. Many changes occur with aging. Among the most important are changes in immune reactivity associated with cell differentiation stages and the phenomenon of inflammaging, understood as subclinical low-grade inflammation, manifested by elevated levels of proinflammatory factors, being both these processes driven by chronic antigen stimulation [129]. 

Inflammaging is considered the basis of most age-related diseases (ARDs). Increased levels of cytokines such as IL-1,2,6,12,15,18,22,23, TNF-α, and INF have been detected in patients affected by many ARDs, such as obesity, metabolic syndrome, diabetes, cardiovascular diseases, and Alzheimer’s disease, together with a decrease of anti-inflammatory factors such as IL1-Ra, IL-4, IL-10, and TGF-b [130].

Release of these cytokines is primarily induced by chronic antigenic stimulation, and sustained by the hyperproduction of ROS, also elicited by the inflammatory response to the antigenic stimuli. On the other hand, the antioxidant system may be depleted in a setting of chronic inflammation, resulting in an imbalance of the redox status and prolonged oxidative stress [131] (Figure 2).

In this vicious cycle, pathophysiological changes, tissue injury, and healing proceed simultaneously. Irreversible cellular and molecular damage that is not clinically evident slowly accumulates over decades, eventually resulting in cutaneous aging and ARDs [132]. 

Long-lived people, especially centenarians, seem to cope with chronic subclinical inflammation through an anti-inflammatory response, called therefore “anti-inflammaging” [133]. On the basis of these observations, efforts have been recently made in order to identify molecules that can improve our response to subclinical inflammation and prevent the consequent cellular damages.

Due to its known anti-inflammatory and antioxidant effects, potential topical and systemic use of curcumin in the treatment and prevention of skin aging has been examined, especially when related to sun exposure (photoaging) [4].

A clinical study on 28 women in their 30s investigated the use of an herbal combination gel containing turmeric, rosemary, and gotu kola (Tricutan^®^) in improving signs of photoaging, reporting a significant improvement in skin firmness and improvement in subjects’ overall self-evaluations after 4 weeks of daily use [134].

A randomized, double-blind, placebo-controlled trial on 47 healthy subjects receiving daily hot water extract of *Curcuma longa* reported a significant inhibition of the increasing in ultraviolet B-induced TNF-α and IL-1β at the mRNA and protein levels compared to placebo. Moreover, the administration of the compound resulted in a significant increase in hyaluronan production from non-stimulated keratinocytes and in a subsequent increase in the water content in facial skin. Besides confirming its anti-inflammatory effects, these results suggest that curcumin may represent an effective moisturizing agent [135].

The effects of curcumin on collagen synthesis discussed above (see Section 1.6) are of definite interest regarding the tone and appearance of facial skin.

Randomized, double-blind, placebo-controlled studies on a larger number of patients are warranted in order to further investigate the possible application of different curcumin formulations in the treatment of skin aging, and whether other ADRs may benefit from its administration. 

### 1.8. Curcumin for the Treatment of Skin Cancer

Non-melanoma skin cancer (NMSC) is the most common cancer in humans, including squamous and basal cell carcinoma (SCC and BCC). Although actinic keratoses (AKs) are lesions characterized by a milder degree of dysplasia, they have up to a 20% risk of progression to squamous cell carcinoma, with eradication being mandatory in affected patients. Mortality from NMSC is low, however, its incidence is high, resulting in a significant public health burden. This makes NMSC a suitable target for chemoprevention and long-lasting research. The skin of the head and neck accounts for 70%–80% of skin cancer cases, chronic sun exposure being a major risk factor for the development of NMSC [136].

Carcinogenesis is a dynamic process that may be divided into two stages: initiation and promotion. The promotion phase is temporally prolonged and potentially reversible, being the target of chemopreventive agents that may prevent the development of an invasive tumor [137].

The pro-inflammatory microenvironment in which cancer develops, and that the cancer itself contributes to produce and maintain, has raised great interest in recent years, representing a potential target for both cancer prevention and treatment. 

Several studies have highlighted that the cyclooxygenases-1 and -2 enzymes (COX-1 and COX-2), induced by UV and other factors, play a significant role in tumor proliferation [138]. In particular, up-regulation of COX-2 induces arachidonic acid metabolism resulting in overproduction of prostaglandin (PG), which directly influence cell growth after binding to specific cell surface receptors, including PG E, F, and I classes of receptors [139,140]. Up-regulation of both COX-1 and COX-2 induces vascular epidermal growth factor (VEGF) production, resulting in angiogenesis and tumor proliferation [141]. Increased levels of prostaglandin are also induced by the down-regulation of tumor suppressor gene 15-hydroxy-prostaglandin dehydrogenase (15-PGDH) [142].

Topical non-steroidal anti-inflammatory drugs (NSAID) represent effective and well-tolerated treatment options for AKs, as they work as nonspecific COX inhibitors. Our group previously demonstrated that local treatment with piroxicam, a NSAID which is active on both COX-1 and COX-2, is a safe and effective agent in the treatment of AKs and field of cancerization, as it blocks the biosynthesis of PGs and in 15-PGDH increased expression [143].

Curcumin selectively inhibits COX-2 in a dose and time-dependent manner [144]. Curcumin may exert this effect by directly targeting COX-2 and PG production and by up-regulating AMP-activated protein kinases (AMPK), that leads to a suppression of COX-2 production [83]. Moreover, curcumin can also prevent biosynthesis of prostaglandin E2 (PGE2) from prostaglandin H2 (PGH2) [145].

Cancer can be considered as the result of a disruption in the physiological balance between cellular proliferation and senescence or apoptosis, regulated by the expression of oncogenes and onco-suppressor genes. Pre-clinical studies pointed out that curcumin induces apoptosis in cancer cells by acting on several pathways. In fact, curcumin showed to induce apoptosis via activation of p53, one of the most studied tumor suppressor genes [8]. Furthermore, curcumin acts on the PI3K/AKT/mTOR pathway, causing a remarkable up-regulation of PTEN, which is a tumor suppressor gene mutated in many types of cancer, and inhibiting the PI3K/AKT axis, which promotes growth and proliferation over differentiation [8]. Of note, curcumin prevents the activation of NF-κB, a protein complex that plays a central role in the survival and resistance of cancer cells [146].

Interestingly, curcumin also showed to induce apoptosis in cancer cells via accumulation of ceramide, a bioactive lipid implicated in apoptosis, cell differentiation, senescence, migration, and adhesion [147]. Moreover, it is able to induce overexpression of TRAIL, one of the most important apoptosis receptors. TRAIL is up-regulated also by the inhibition of NF-kB, that may in turn be induced by curcumin [148]. Curcumin showed to induce apoptosis in a concentration-dependent manner in human colon cancer, human myelocytic leukemia, human choriocarcinoma, and melanoma cells through the activation of c-Jun N-terminal kinases (JNKs), a group of mitogen-activated protein kinases involved in redox reactions and apoptosis induction [149,150,151,152,153]. It has been suggested that up-regulation of JNK by curcumin may enhance the therapeutic efficiency of chemotherapy drugs, but the real benefits of a combination therapy are still a matter of debate [154,155,156].

In spite of its incidence being low, melanoma is the foremost aggressive kind of cutaneous cancer, being extremely resistant to chemotherapy and radiotherapy. To date, uncontrolled sun exposure is regarded as the only modifiable risk factor for melanoma [4]. Although no solid evidence regarding the role of specific nutrients in the prevention of melanoma is available to date, a possible link between diet quality and melanoma risk has been postulated [157]. Several in vitro studies assessed the effects of curcumin on melanoma cells proliferation and viability [158,159,160]. Of note, curcumin showed to affect the growth of melanoma cells selectively, through all the above-mentioned mechanisms that result in the induction of the apoptotic process. Moreover, curcumin could be able to arrest cell cycle in G2/M by directly inhibiting cyclic nucleotide phosphodiesterases (PDEs) [161]. 

Studies on animal models largely confirmed the effects of curcumin against melanoma, especially when administrated in formulations that ameliorate curcumin bioavailability, such as nanocapsules [20]. As mentioned above, attempts to associate curcumin with conventional drugs in order to potentiate their efficacy on melanoma have been made, with promising results [162,163,164,165]. However, results observed in preclinical models may not be mirrored in clinical studies, due to problems related to the in vivo low bioavailability and metabolism of curcumin, or to the significant differences existing between tumors generated in animal models and human cancer. Bypass of these limitations represents a very promising field for future applicative research, paving the way to clinical trials.

### 1.9. Curcumin for the Treatment of Skin Infections

The efficacy of curcumin to control skin infection diseases was also investigated both in vitro and in vivo in animal models. Cutaneous infections may be caused by a wide variety of microorganisms including bacteria, fungi, viruses, and parasites. The most common bacteria responsible for this illness belong to the genera *Corynebacteria*, *Propionibacteria*, and *Staphylococci*. These microorganisms, which normally live on the skin as commensals, playing a crucial role in the maintenance of skin homeostasis, may also cause cutaneous infections, acting as opportunistic pathogens [166]. Among *Staphylococcus* spp., *Staphylococcus aureus* is responsible for a wide spectrum of skin infections such as boils, impetigo, cellulitis, and folliculitis. *Staphylococcus epidermidis* and *Propionibacterium acnes* are also part of the human skin microbiota and both play a direct role in the development of acne vulgaris. In most cases primary skin infections are not invasive diseases in immunocompetent individuals. However, because of the increasing number of microorganisms resistant to multiple drugs, skin bacterial infections can remain extremely difficult to treat. Some staphylococcical strains have developed resistance to both naturally and semisynthetic beta-lactamase-resistant penicillins (i.e., oxacillin, methicillin, and dicloxacillin). *Propionibacterium acnes* is naturally resistant to some antibiotics such as 5-nitroimidazole, aminoglycosides, sulfonamides, and mupirocin, although it is generally susceptible to a numerous type of antibiotic drugs. Over the last years resistance of *Propionibacterium acnes* to antibiotic therapies has also gradually increased becoming a worldwide concern, with maximal resistance for erythromycin and clindamycin and less frequent resistances to tetracycline, in parallel with the most common topical administration of macrolides [167,168,169]. Beside the emergence of acquired resistance in bacteria against the current antibiotics used in clinical setting, another main concern is the overall variation of the human skin microbiota, related to the emergence of resistant microbial species induced by the selective pressure exerted by antibiotic agents [170]. This issue should limit topical and/or systemic antibiotics therapies for long term in the management of skin diseases such as acne vulgaris. Thus, novel therapeutic approaches are required to treat skin infectious diseases. In the last years researchers have focused their attention on the development on plant derived natural products, as alternative or complementary option to traditional medicine. Indeed, the bioactive aromatic compounds obtained from some of the medicinal herbs have been shown to possess potential antimicrobial properties. In this scenario, the antimicrobial activity of curcumin has been extensively investigated due to its large uses and safety profile even at high doses tested in clinical trials [170]. In vitro studies demonstrated that *S. aureus* is one of the Gram-positive strains susceptible to the inhibitory effect of curcumin. Further, the curcumin efficacy has also been shown against methicillin-resistant *S. aureus* (MRSA) either alone or in association with conventional antibiotics [170]. A significant dose dependent microbicidal activity in vitro against both *S. aureus* and *P. acnes* was obtained by blue light activated curcumin. This microbicidal property of light irradiated curcumin could be attributed to the bacterial cell membrane disruption mediated by vanillin, a curcumin photolytic degradation product [171]. Additionally, as demonstrated for the first time by Almeida et al., curcumin, acting as a photosensitizer, enhanced the bactericidal effect of photodynamic therapy against MSRA in a murine model of intradermal infections [172]. As the blue light safety profile in mammalian cells has been proven, the photolytic treatment of curcumin could be used in the future to eradicate bacterial skin infections caused by multi drug resistant strains of *S. aureus* and *P. acnes*. Moreover, an improved antibacterial activity in vitro against both macrolide-sensitive and resistant strains of *P. acnes* was also obtained by liposomal gel formulations containing curcumin combined with lauric acid [33]. These results were supported by preclinical studies showing that curcumin co-applied with lauric acid in liposomal gel, in a rat model of acne vulgaris, significantly reduced the comedones count and the inflammatory cytokine production such as TNF-alpha and IL-1-beta [34]. Additional in vivo studies showed that myristic acid acts in synergistic way with curcumin, loaded in the microemulsion carrier, in inhibiting *S. epidermidis* growth. These results suggest the potential use of curcumin-loaded microemulsions as alternative therapy in *S. epidermidis*-associated diseases like acne vulgaris [34]. Beside to bacteria, several genera of fungi may be responsible for superficial and cutaneous mycoses. Particularly, dermatophytes represent the most common fungal pathogens involved in skin infections. Among dermatophytic fungal pathogens, *Trychophyton rubrum* has become the most frequent species worldwide, causing mainly tinea pedis and tinea unguium [173]. Over the last years, like bacteria, fungi have also been developing resistance to conventional antimycotic drugs. In addition, due to low number and toxicity of the antifungal agents currently in use, the treatment of skin mycotic infections is often difficult. Therefore, there is an urgent necessity to develop novel antifungal molecules able to target specific cellular and or molecular mechanisms involved in fungal pathogenicity, to control these illnesses. In this context, curcumin encapsulated in nanoparticles administered after photodynamic therapy has been shown to completely inhibit the growth of *T. rubrum* in vitro, through the release of reactive oxygen (ROS) and nitrogen species (RNS), which play an important role in inducing fungal death by apoptosis [174]. Altogether this experimental evidence suggests that curcumin alone or combined with phototherapy may be a potential and very promising candidate in treating bacterial and fungal skin diseases, overcoming the multi-drug resistance of pathogens.

### 1.10. Molecular Docking Analysis Highlights the Role of Curcumin in the Control of Skin Disorders

Molecular docking is a computational tool able to predict the binding mode of a ligand with a protein of known three-dimensional structure. Docking can be used to perform virtual screening on large libraries of compounds, rank the results, and propose structural hypotheses of how the ligands inhibit the target, which is invaluable in the research of novel inhibitory compounds [175]. A variety of molecular docking studies have been applied to demonstrate the role of the curcumin molecule in targeting a selection of proteins actively involved in various pathologies unrelated to skin disorders. A molecular docking analysis has been performed on diketone form of curcumin molecule with acetylcholinesterase (AChE), indicating that this molecule exhibits a large binding affinity, and suggesting the use of curcumin to inhibit AChE and balance the level of acetylcholine as an alternative to the present Alzheimer’s disease treatments [176]. Focusing on the effects of phytochemicals on some important ocular disorders (Eales, Diabetic Retinopathy, Uveitis, Age related Macular Disorder, Central Retinal Vein Occlusion), virtual screenings identified the potentiality of ginkgolide, D-pinitol, gugglesterones, berberine, and curcumin molecules against the above-mentioned ocular disorders [177]. Curcumin analogues have also been evaluated for COX-2 inhibition and anti-inflammatory activity. Molecular docking studies show that these designed analogues significantly enhance their COX-2 selectivity, suggesting the route to the design of novel inhibitors [178]. Subsequently, molecular docking was carried out to evaluate the binding efficiency of curcumin with peroxisome proliferator-activated receptor gamma (PPARγ). The experimentally validated results demonstrate a preventive role of curcumin on diet induced insulin resistance in rats by ameliorating the altered levels of metabolic changes [179]. Molecular docking was also performed to evaluate the interaction of curcumin with JAK2, an important upstream kinase that phosphorylates STAT3. The obtained results, supported by experimental evidence, indicate that curcumin is able to exert anti-tumor activity through the inhibition of the STAT3 signaling pathway [180]. Recently a study aimed at understanding the binding of curcumin and its analogues to different PDE-4 subtypes, has been carried out. Docking analysis has been employed to design curcumin derivatives with increased anti-inflammatory activity [181]. Concluding, several studies demonstrate that the activity of Sortase A, a bacterial surface protein from *S. aureus* and *Streptococcus mutans*, can be inhibited by curcumin and its analogues [182,183,184]. 

Molecular docking studies describing the interaction of curcumin with molecular targets involved in the development of skin disorders are nowadays not available in the literature. To overcome this limitation, we used protein-ligand molecular docking to evaluate binding mode and interaction energy of the curcumin towards six major enzymatic targets, indicated in this review as responsible for most of skin disorders. The docking simulations were executed using the AutoDock Vina 1.1.2 program, through the AutoDock/Vina PyMOL plugin (http://wwwuser.gwdg.de/~dseelig/adplugin.html) [185,186]. The curcumin 3D structure (Figure 3), in the form of an SDF file, was obtained from the PubChem compound database (https://pubchem.ncbi.nlm.nih.gov, compound CID: 969516). 

Crystal structures of nucleotide phosphodiesterases-1 (PDE-1; PDB ID: 4NPW), protein kinase B (AKT; PDB ID: 6HHF), protein kinase C (PKC) theta (PKCθ; PDB ID: 5F9E), serine/threonine-specific protein kinase (PhK; PDB ID: 2Y7J), cyclooxygenase-2 (COX-2; PDB ID: 5F1A), and phosphoinositol-3-kinase (PI3K; PDB ID: 4WAF), have been used as receptors for the molecular docking simulations [187,188,189,190,191,192]. Each chosen structure displays a co-crystallized compound (Table 3), which was re-docked as a test using the same simulation parameters. The side chains belonging to the active sites were considered rotatable to improve mobility of the receptors during the simulations. The dimensions of the docking box were tailored depending on the active site typology and structure. The AutoDock/Vina program selects, for each docking simulation, 10 ligand poses representing the cluster centroids of all the evaluated solutions. Each docking simulation run takes about 20’ (elapsed real time) on a dedicated AMD Ryzen 7 1700X CPU workstation.

Curcumin was docked in the active site of six enzymes (i.e., PDE1, AKT, PKCΘ, PhK, COX-2 and PI3K), which are involved in several patterns described in this review. In all these receptors the molecule shows interaction energies that ranges between −10.0 and −8.0 kcal/mol, suggesting a possible inhibitory role (Table 1). Moreover, when the co-crystallized inhibitors detected in the PDB files were re-docked using the same simulation conditions, they display energies that are comparable with those evaluated for the curcumin. In particular, the curcumin molecule docked in the active sites is fully stabilized by several hydrophobic contacts and hydrogen bonds established with the active site residues (Figure 4 and Figure 5).

Chemically, curcumin is a diarylheptanoid (IUPAC name: (1E,6E)-1,7-Bis(4-hydroxy-3-methoxyphenyl) hepta-1,6-diene-3,5-dione), a tautomeric compound existing in enolic form in organic solvents or as a keto form in water. The two halves of this highly symmetric compound are able to dispose in a wide range of conformations due to the rotatable bonds located in the center of the molecule (Figure 3). As a consequence, curcumin is able to arrange the two substituted phenol rings (i.e., the 4-hydroxy-3-methoxyphenyl) to establish both hydrophilic and hydrophobic interactions. These results show that the curcumin has a high ligand-potentiality for a wide range of macromolecules, making us hypothesize that it may interact with many other enzymes or proteins which were not considered in this simulation analysis.

## 2. Conclusions

Turmeric is a plant known by its medicinal use, dating back to 4000 years ago in the Vedic culture in India and is widely used in herbal and complementary medicine.

A growing amount of evidence confirms that curcumin might modulate those phenomena involved in inflammatory, proliferative, and infectious disorders of the skin.

To current knowledge, curcumin is a low-cost, well-tolerated agent. However, due to the functional pleiotropy of this molecule resulting in a large spectrum of actions that are still not fully understood, surveillance is advisable, especially in its over-the-counter use. The simulation results confirm the large conformational adaptability of the curcumin compound, indicating a wide range of unknown possible interactors and suggesting the route for the discovery of new targets.

Bypass of limitations related to curcumin in vivo use, such as low oral bioavailability and metabolism, and larger experience with its intravenous administration, would pave the way to larger clinical studies that could provide clinicians solid data regarding curcumin safety and the possible clinical benefits of curcumin-containing products to skin health. The possible use of curcumin in combination with traditional drugs and the formulations of novel delivery systems represent a very promising field for future applicative research.

## Figures and Tables

**Figure 1 nutrients-11-02169-f001:**
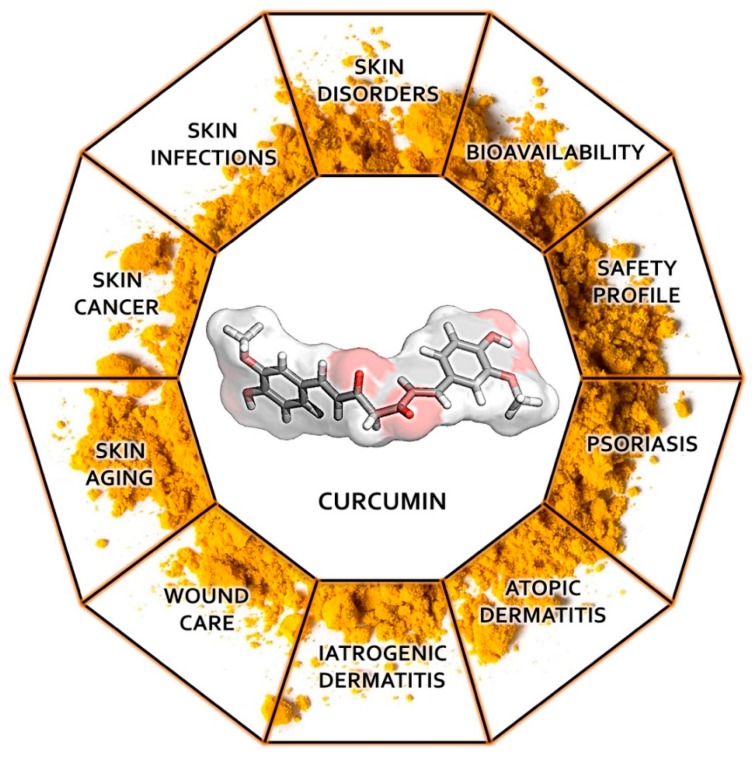
Graphical abstract.

**Figure 2 nutrients-11-02169-f002:**
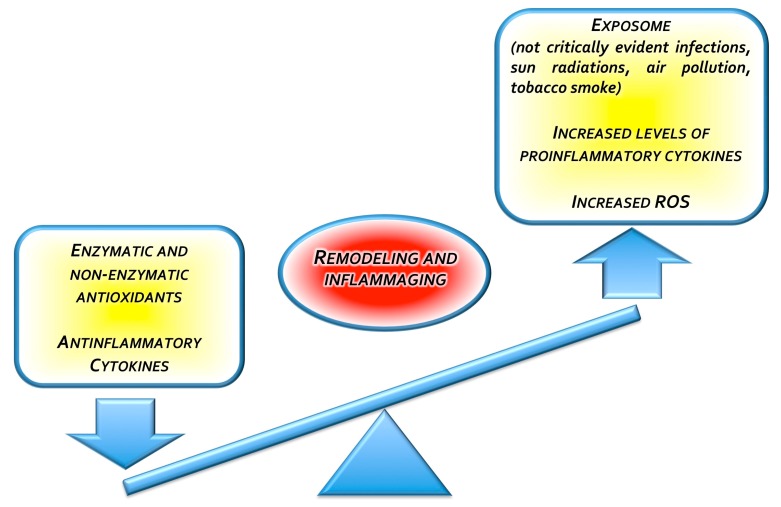
Inflammatory status imbalance leading to inflammaging.

**Figure 3 nutrients-11-02169-f003:**
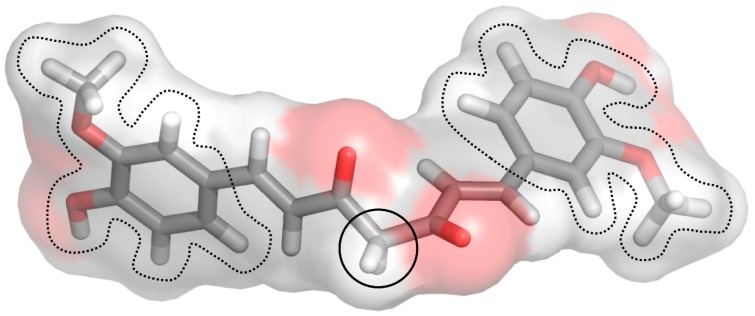
Stick representation of the keto form of the curcumin molecule. The red, grey, and white colors indicate the oxygen, carbon, and hydrogen atoms, respectively. A black circle indicates the center of the symmetric molecule, while the 4-hydroxy-3-methoxyphenyl, present in each of the two compound halves, is enclosed by a dotted line. This image was generated using the program PyMOL (The PyMOL Molecular Graphics System, Version 2.0 Schrödinger, LLC, New York, NY, USA).

**Figure 4 nutrients-11-02169-f004:**
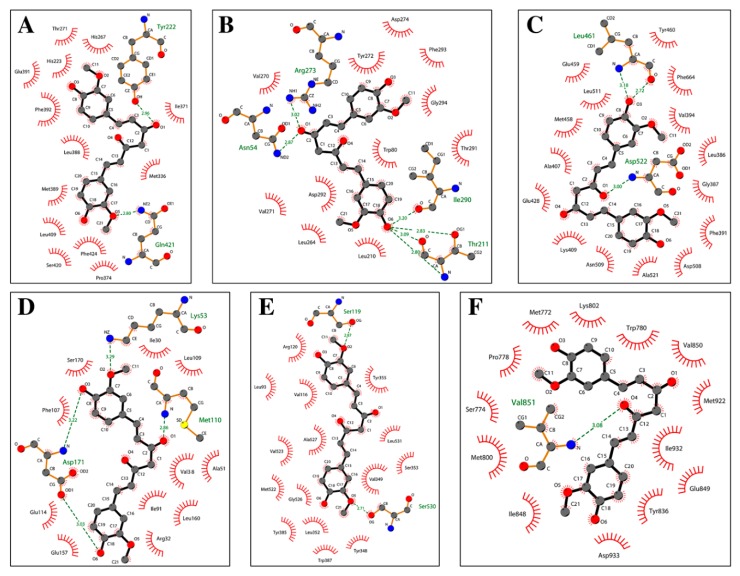
Schematic view of the best molecular docking complexes between curcumin and (**A**) PDE1 (4NPW), (**B**) AKT (6HHF), (**C**) PKCΘ (5F9E), (**D**) PhK (2Y7J), (**E**) COX-2 (5F1A), and (**F**) PI3K (4WAF). The residues interacting through hydrogen bonds (green dashed lines) are shown in ball-and-stick, while the residues in contact with the ligand are indicated by circle sections with rays. This was produced using the LigPlot+ software (Laskowski R.A., Swindells M.B. LigPlot+: multiple ligand-protein interaction diagrams for drug discovery. (2011) J. Chem. Inf. Model. 51, 2778–2786.).

**Figure 5 nutrients-11-02169-f005:**
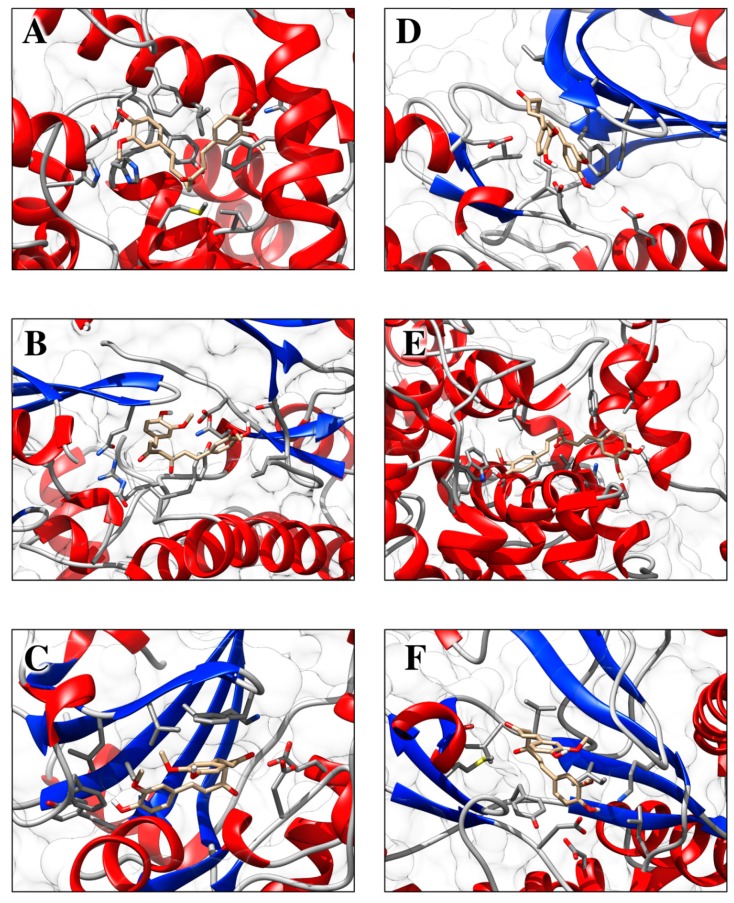
Molecular view of best docking complexes between curcumin and (**A**) PDE1 (4NPW), (**B**) AKT (6HHF), (**C**) PKCΘ (5F9E), (**D**) PhK (2Y7J), (**E**) COX-2 (5F1A), and (**F**) PI3K (4WAF). The β-strands are represented by blue arrows, while the α-helices and the loops are shown as red spirals and light grey wires, respectively. The curcumin, hosted in the active site, is indicated by stick model colored by atom type. This picture was generated using the program Chimera (Pettersen E.F., Goddard T.D., Huang C.C., Couch G. S., Greenblatt D.M., Meng E.C. and Ferrin T.E. (2004) UCSF Chimera—A visualization system for exploratory research and analysis. J. Comput. Chem. 25, 1605–1612.).

**Table 1 nutrients-11-02169-t001:** Formulations of curcumin for oral, topical, or intravenous use investigated in preclinical and clinical studies for enhanced bioavailability listed in this review.

Route of Administration	Formulation	Reference
Oral	Curcumin-piperine nanoparticles	[6]
	Curcumin-loaded PLGA nanoparticles	[13]
	CE-complexed curcumin	[14]
	Curcumin-loaded self-nanomicellizing solid dispersion based on RA (RA-Cur)	[15]
	Colloidal Submicron-Particle Curcumin (Theracurmin^®^)	[16]
	Curcumin-loaded liposomes	[6,17]
	Curcumin micelles	[17,18]
	Lecithin-based formulation (Meriva^®^)	[19]
	Curcumin nanocapsules	[20]
Topical	Curcumin-loaded chitosan-alginate sponges	[21]
	Curcumin-loaded oleic acid-based polymeric bandages	[22]
	Curcumin-loaded alginate foams	[23]
	Curcumin-incorporated collagen films	[24]
	Hydrogel system containing curcumin micelles	[25]
	Curcumin nano-emulsion	[26]
	Curcumin-β-Cyclodextrin nanoparticles	[27]
	Chrysin-curcumin-loaded nanofibers	[28]
	Curcumin-loaded transdermal patches	[29]
	Curcumin nanoparticles	[30,31,32]
	Curcumin-loaded-liposomes	[30,33,34]
	Curcumin-loaded chitosan nanoparticles impregnated into collagen-alginate scaffolds	[35]
Intravenous	Curcumin-loaded solid lipid nanoparticles	[36]
	Curcumin-loaded liposomes	[37,38,39]

PLGA, polylactic-co-glycolic acid; CE, cyclodextrin; RA, rebaudioside A.

**Table 2 nutrients-11-02169-t002:** Ongoing clinical trials with curcumin in skin disorders.

Major Outcome Measures	Pain Intensity Measured by Visual Analog Scale (VAS)	Change in Erythema 1 Day After UV Exposure Change in Erythema 2 Days After UV Exposure Change in Erythema 1 Day After UV Exposure
Study design	Randomized, double-blind, Phase 1 clinical trial	Randomized, double-blind
Intervention model	Parallel assignment	Parallel assignment
Topical or ingested curcumin containing product	Topical	Ingested
Intervention/treatment	Drug: Triamcinolone Drug: Turmeric paste	Dietary supplement: Crucera-SGSDietary Supplement: Meriva 500-SF
ClinicalTrials.gov Identifier	NCT03877679	NCT03289832
Study	The Effect of Topical Curcumin Versus Topical Corticosteroid on Management of Oral Lichen Planus Patients	Effect of Orally Delivered Phytochemicals on Aging and Inflammation in the Skin
Condition or disease	Oral lichen planus	UV-induced skin erythema

**Table 3 nutrients-11-02169-t003:** Molecular docking simulations results.

Receptor Name (PDB ID)	Co-CrystallizedCompound Name	Co-CrystallizedCompound Structure	Co-CrystallizedCompound Docking EnergyΔG(kcal/mol)	Curcumin Docking EnergyΔG (kcal/mol)
PDE1 (4NPW)	Inhibitor 19A((7,8-dimethoxy-N-[(2S)-1-(3-methyl-1H-pyrazol-5-yl)propan-2-yl]quinazolin-4-amine))	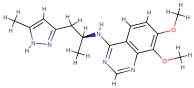	−9.4	−10.1
AKT (6HHF)	Borussertib	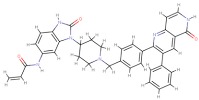	−14.5	−9.8
PKCΘ (5F9E)	1-Benzyl-2,2-dimethyl-7-(2-oxo-3H-imidazo[4,5-b]pyridin-1-yl)-3H-quinazolin-4-one	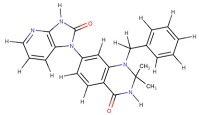	−11.9	−9.6
PhK (2Y7J)	Sunitinib	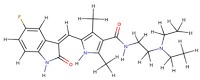	−8.4	−8.4
COX-2 (5F1A)	Salicylate	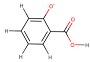	−6.6	−8.4
PI3K (4WAF)	Tetrahydropyrazolo [1,5-A] pyrazine	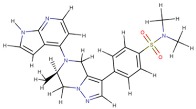	−9.2	−7.6

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
