# Peer review of "Potential of Curcumin in Skin Disorders"

_nutrients, 2019, doi:10.3390/nu11092169_

Round 1
Reviewer 1 Report
This is a comprehensive review of the evidence for protective and restorative effects of curcumin treatments in skin health and disease. This topic is very timely with many publications surrounding the beneficial health effects of curcumin published this year in a wide variety of journals. The article covers a wide range of skin conditions and details the potential biochemical actions/mechanisms as well as the clinical evidence for its efficacious use against skin disease.
Overall comments:
1) I think there is scope for more critical analysis of the available studies within the article with little mention of studies which have not shown significant beneficial effects and little mention of the limitations of many of the clinical studies until the very end of the article.
2) The authors also present original data on the binding actions of curcumin to a number of key enzymes reported in the literature to be a target of curcumins therapeutic actions. While this data is interesting and highlights the potential wide ranging physiological effects of curcumin (potentially positive and negative), the review does not flow well into this section and it feels disjointed in structure. This second part need to be reworked in order that the article doesn't lose its flow and that the importance of the data presented is explicitly obvious to the reader who may not be a biochemist/computational biologist. While I understand the need to describe what you have done there is too much jargon in this section.
3) There appears to be a lot of publishing activity on this topic so worthwhile rechecking the literature to ensure the most up to date studies/evidence is included.
Below are some minor comments:
P1, L42: You reference a relatively recent systematic review on curcumin and skin health at the beginning of your article without really referencing that it was a systematic review. Can you clarify how your literature review is furthering the field following on from this article?
P2,L592: It is not clear what you mean when you say that 'in addition bioavailability is low' when you have already discussed this in the previous sentences. Do you mean to say bioactivity is low?
P2,L67: Can you comment on any longer term studies looking at systemic/topical curcumin use? Many of the studies only use curcumin for a few months. Curcumin supplements are widely available to the general public over the counter and with the publicity surrounding its potential beneficial health effects they appear to be becoming increasingly popular. I think it is important to comment briefly on this given that we don't know the long term effects. While the doses recommended for over the counter curcumin are generally lower than in the clinical studies this raises other questions about the bioavailability of these supplements and the fact that people could be taking them when there is no evidence that there is clinical benefit- perhaps more of an ethical rather than a medical consideration at this time which doesn't just relate to curcumin, however, I think it is interesting and important point to consider.
P3, L109: It is unclear on first read what you mean by ligand here and therefore unclear what mechanism you are trying to present.
P3, L117: Define PhK and not enough context on the model/experimental setting without going to the reference. Also in this section you mention side effects in AD studies- where there any in psoriasis studies?
P4, L166: Define iatrogenic dermatitis as non clinicians may not be familiar with this terminology.
P5 Table 1: Consider adding references to the table.
P8, L329-336: Could COX inhibition be responsible for gastrointestinal side effects due to inhibition of PG synthesis? I think more commentary on the possible downsides of the wide ranging effects of curcumin should feature within the article to create a more balanced review rather than just mentioning it at the end.
Reviewer 2 Report
The article is a good compilation of the use of Curcumin in treatment of skin disorders.
However, I would suggest the authors to consider including infographic representation of the targets and effects which will increase the value of this manuscript. The amount of text can be reduced to tables which will give easy understanding to the readers.
Listing studies under each section based on the models used would be good.
It will also be good if information on the interaction of Curcumin with other commonly used chemicals in skin treatment.
May require a English check to make sentences complete.
Reviewer 3 Report
Curcumin is widely considered as a therapeutic in skin diseases – the “curcumin and skin disease” entry in PubMed returns 595 results. There are also many reviews on this subject published in 2018-19. Therefore, another curcumin-subjected review should contains new aspects justifying its publication. I think that this manuscript fulfills such requirement as it links a review of preclinical and clinical studies on curcumin with an original computer-based modeling of the interaction between curcumin and its potential targets in skin diseases. However, there are many papers on molecular docking studies with curcumin and the authors should underline, in 1-2 sentences in the Introduction section, novelty aspects of their review (just update?) and molecular studies.
The main concern with this manuscript that it mostly describes in vitro effects of curcumin and points at its low bioavailability. Therefore, the title “Potential of curcumin in skin diseases” would better reflect the content of the manuscript.
Detailed
Bioavailability of curcumin
This section should be rewritten through the introduction of information on formulation of curcumin increasing its bioavailability (nutraceuticals). In its present form this section suggests that orally administrated curcumin is not bioavailable, but next sections present some significant results induced by curcumin administrated in that way.
Expression such as: “only traces”, “nanomolar levels” say little, if any, about bioavailability, which is a fraction of administrated curcumin that reaches the circulation.
How is curcumin transported from the intestine to blood? Is curcumin applied intravenously in clinical trials?
Again, “low levels” (rows 71-72) says little
Curcumin for the treatment of skin cancer
Row 339 - senescence (apoptosis) – is misleading – these two distinct phenomena (processes) should not be mixed, so “senescence or apoptosis” would be better
Figure 1 – I suggest to remove the table as it exactly repeats the main text and instead to present some formulations of curcumin used in preclinical and clinical studies.
Conclusions
Results of molecular modelling should be linked with curcumin bioavailability.
Does dietary curcumin matter in the prevention of skin diseases?
What are perspectives of therapeutic intravenous administration of curcumin?
